# Scalable Multi-phase Word Embedding Using Conjunctive Propositional Clauses

## Abstract

The Tsetlin Machine (TM) architecture has recently demonstrated effectiveness in Machine Learning (ML), particularly within Natural Language Processing (NLP). It has been utilized to construct word embedding using conjunctive propositional clauses, thereby significantly enhancing our understanding and interpretation of machine-derived decisions. The previous approach performed the word embedding over a sequence of input words to consolidate the information into a cohesive and unified representation. However, that approach encounters scalability challenges as the input size increases. In this study, we introduce a novel approach incorporating two-phase training to discover contextual embeddings of input sequences. Specifically, this method encapsulates the knowledge for each input word within the dataset's vocabulary, subsequently constructing embeddings for a sequence of input words utilizing the extracted knowledge. This technique not only facilitates the design of a scalable model but also preserves interpretability. Our experimental findings revealed that the proposed method yields competitive performance compared to the previous approaches, demonstrating promising results in contrast to human-generated benchmarks. Furthermore, we applied the proposed approach to sentiment analysis on the IMDB dataset, where the TM embedding and the TM classifier, along with other interpretable classifiers, offered a transparent end-to-end solution with competitive performance.

## 1 Introduction

Recent advancements in the development of language processing applications have significantly propelled the field of Artificial Intelligence (AI). A pivotal innovation in this domain is the application of Large Language Models (LLMs), which primarily utilize embeddings during the early stages of the model development. During training, dense vectors are extracted for each word in a space that conveys the word's context and location from the training dataset. These embeddings are then leveraged across various architectures to construct diverse applications.

A novel methodology in this context involves enhancing ML through the use of logical propositions Granmo (2018). This approach enables the representation of a target word by a set of words crucial in shaping its meaning. One key distinction of embeddings generated using logical propositions, as opposed to traditional Deep Learning (DL) methods, is their interpretability. The output, represented by clauses, can be understood and analyzed in line with the original logical framework. In previous work Bhattarai et al. (2024), embeddings were derived for a vector of target words. That method produces a fused output that obscures the individual contributions of each input word, thus forfeiting the interpretability advantage inherent in the original TM algorithm.

An experiment utilizing the One Billion Word dataset Chelba et al. (2013) and the Tsetlin Machine Auto-Encoder (TM-AE) model Bhattarai et al. (2024) revealed that training the model on 100 input target words required approximately 9 hours and 35 minutes. Notably, the embedding generated during each training session is specific to the particular set of input target words and cannot be reused in scenarios involving the addition, removal, or substitution of any element within the input array. Consequently, the time-intensive training process is bound to the initial input string, limiting its reusability in other applications. This scenario highlights the scalability challenges associated with the TM-AE model Bhattarai et al. (2024) when applied to downstream tasks, underscoring the need for further optimization and improvements to enhance its efficiency.

This research aims to introduce a novel approach for scalable knowledge extraction from any word within the training vocabulary. The objectives and contributions are threefold:

1. To collect and encapsulate the knowledge for each target word in the dataset's vocabulary in a manner that facilitates the construction of a scalable model for downstream tasks.

2. To build embeddings for a vector of target words using the extracted knowledge while preserving the interpretability properties of the original TM algorithm.

3. To apply these embeddings in data augmentation, enabling the evaluation of embedding quality by testing on unseen data and analyzing the effect of variations in the trained dataset, thereby enhancing model classification robustness.

## 2 BACKGROUND AND RELATED WORK

In the realm of AI applications, NLP has witnessed remarkable advancements over the past decade. Prominent algorithms such as Word2Vec and GloVe have played pivotal roles in this progress Goldberg & Levy (2014); Pennington et al. (2014). Despite the promising results achieved by these embedding algorithms, there remains a pressing need for further improvements. For instance, FastText focuses on subword information, thereby enhancing its capability to handle rare words and misspellings Bojanowski et al. (2017), while ELMo generates contextualized word embeddings by considering the entire sentence Peters et al. (2018).

A novel approach to representing ML through a logical structure is the TM, which emphasizes transparency and the ability to provide justifications for outcomes. TM have seen significant advancements through various contributions. Maheshwari et al. (2023) introduced REDRESS, a method for generating compressed models tailored for edge inference using TM. Sharma et al. (2023) enhanced the robustness and pattern recognition capabilities of TM through the Drop Clause technique. Seraj et al. (2022) explored the application of TM to solve contextual bandit problems. Yadav et al. (2021) demonstrated the human-level interpretability of TM in aspect-based sentiment analysis. Abeyrathna et al. (2021) developed a massively parallel and asynchronous TM architecture for efficient scaling. Additionally, Abeyrathna et al. (2023) proposed a method to build concise logical patterns by constraining the clause size in TM.

In the field of NLP, several models have leveraged TM for various applications. For instance, in the work Yadav et al. (2022), robust and interpretable text classification models were developed by learning logical AND rules with negation. Another study Saha et al. (2023) used TM to discover interpretable rules for tasks such as sentiment analysis, semantic relation categorization, and dialogue-based entity identification. The work Zhang et al. (2023) applied TM in sentiment analysis and spam review detection for Chinese text, aiming to strike a balance between interpretability and accuracy when compared to DL models. Additionally, a TM-AE was designed to identify word embeddings Bhattarai et al. (2024) using the Coalesced Tsetlin Machine (CoTM) structure Glimsdal & Granmo (2021), which incorporates voting on a set of outputs with weighted clauses representing their contributions to the algorithm. That embedding method has demonstrated superior performance compared to DL alternatives.

## 3 METHODOLOGY

In this section, we will present the proposed algorithm. First, we explain the preprocessing of the input data and how CoTM can be employed to form the two phases utilized in this work. Thereafter, we introduce the two-phase architecture based on the TM-AE architecture.

### 3.1 COALESCED TSETLIN MACHINE

The process starts with preparing the input $X$, which is a binary vector encapsulating the context information to be trained on. It is twice the length of the vocabulary ($V$) of the training dataset to accommodate all original features and their negation. These original and negated features are called literals, and the original features are basically the vocabulary in the dataset. The model can be trained using unlabeled document inputs. If the target output is 1, the input $X$ is created from features found in documents containing the target word. If the output is 0, the input $X$ is created

from features found in documents that do not contain the target word. The features mentioned in the documents are activated in $X$ by setting their value to 1, while the rest of the features are set to 0. Negated features are represented by reversed values. It is important to note that the construction of $X$ is independent of the frequency with which a word appears in a given document; its mere presence, even if mentioned only once, is sufficient.

To explain the formulation of the input $X$, let us look at an example. We assume that the vocabulary contains eight words: $word_1$, $word_2$, ..., $word_8$. Suppose we have two documents: $document_1$, which contains the words $[word_3, word_2, word_4]$ and $document_2$, which contains the words $[word_6, word_3]$ and we want to form $X$ for the target word $word_3$. For the first part of $X$, we select the features corresponding to the words in $document_1$ and $document_2$ and set their values to 1. The rest of the features are set to 0. Thus, we have: $X_{original} = [0, 1, 1, 1, 0, 1, 0, 0]$, where $X_{original}$ is a binary vector of length 8. For the second part of $X$, we take the negation of $X_{original}$. This means setting the value of each feature to the opposite of its value in $X_{original}$. Thus, we have: $X_{negation} = [1, 0, 0, 0, 1, 0, 1, 1]$. Finally, we concatenate $X_{original}$ and $X_{negation}$ to form $X$, as $X = Concat[X_{original}, X_{negation}]$, namely, $X = [0, 1, 1, 1, 0, 1, 0, 0, 1, 0, 0, 0, 1, 0, 1, 1]$. This $X$ represents the active literals required to train the TM for the target $word3$, and it has a length of 16 literals. The two phases used in this work are distinguished by the approach employed in forming $X$, to be explained in the next section.

The process of evaluating the clauses begins with the input $X$, where the literals of the input are matched with the propositional logic represented by the conjunctive clauses. These clauses are different forms of description of the target word, composed of literals. The literals memorized in a specific clause are determined during the training process. More specifically, an election process occurs throughout the training period of the model to update the depth of memorizing or forgetting the literals in memory and thus include or exclude them from the clause. Each literal corresponds to a Tsetlin automaton (TA), which decides to memorize or forget the literal based on the reward/penalty dynamics that it receives. In more details, each TA is assigned to a specific literal in $X$ and evaluates its state across a range of $2N$ levels, where level 1 corresponds to the most extensive forgetting and level $2N$ represents the highest degree of memorization. Fig. 1 illustrates the operation of the TM, highlighting how the TA collaborate to evaluate the input $X$, thereby forming the clause. The input $X$ consists of $V$ features, resulting in a total of $2V$ literals. In the figure, both $x_1$ and $\neg x_V$ have successfully progressed beyond the $N$ state, thereby contributing to the formation of the clause, represented by the blue area. For more details on the clause formation, see Granmo (2018).

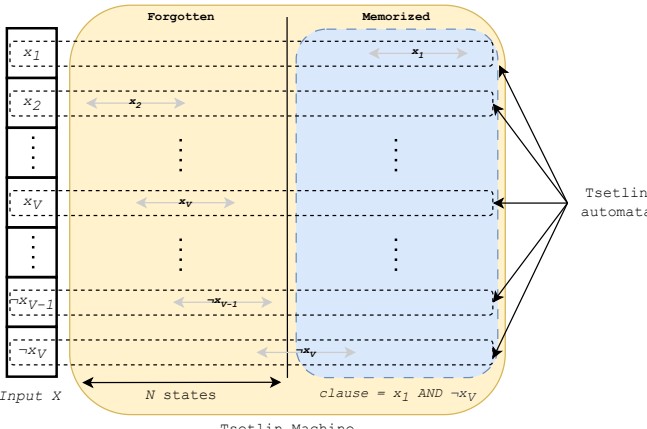

Figure 1: Illustration of Tsetlin automata in a Tsetlin Machine: Demonstrating the processing of input $X$ through multiple states to form a clause.

In the original TM, the output of the evaluation process is subject to a voting sum calculation, which is influenced by the voting margin hyperparameter $T$, ensuring the presence of the required number of supporting clauses for the output. In CoTM, the inference process includes an additional type

of processing that employs weights for the outputs. A dedicated memory space is provided for the weights and for each clause with the number of outputs it infers. Thus, three variables are used to calculate the output (Fig. 2). The proposed output $Y$ from input $X$ is predicted using propositional and linear algebra, as $Y = U \left( W \cdot And \left( C, X \right) \right)^T$. Here $Y$ is the multi-output prediction, $U$ is

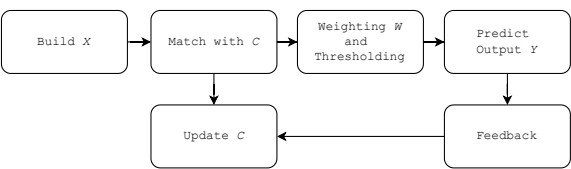

Figure 2: CoTM architecture: Depicting the process flow from input $X$ through matching, updating, weighting, thresholding, and predicting output $Y$ with feedback.

an element-wise unit step thresholding operator, $W$, $C$ and $X$ are the memory arrays of weights, clauses and input $X$ respectively, $And$ is a row-wise AND operator.

Each clause is updated based on the input $X$ through three types of updates Granmo (2018):

1. **Memorization process**: Increase the memorization of true literals in $X$ if the clause matches it and the output result is 1. The literals in $X$ whose value is false are subject to increased forgetting through random selection using the hyperparameter $s$.

2. **Forgetting process**: Increase the forgetting of all true literals in $X$ if it does not match the clause and with a probability that depends on $s$ used to select the literals.

3. **Invalidation**: Increase the memorisation of false literals in $X$ to change the clause to reject true literals in $X$.

## 3.2 TSETLIN MACHINE AUTOENCODER

In the architecture proposed in Bhattarai et al. (2024), the input $X$ is constructed by randomly shuffling a set of target words. The process for preparing the input $X$ is outlined in Algorithm 2 in the appendix. This algorithm involves combining input documents, where the model receives the documents in their vectorized form, with the vocabulary serving as a reference to track the position of each word within the encoding throughout the training process.

The TM-AE processes a sequence of input target words, aggregating them without repetition to form the basis for training and determining their embeddings. This type of embedding integrates the contextual information of each word in the input with the other words. Given the random mixing and the number of examples $r$ applied in each epoch, each word has an equal probability of being at position $j$ in the input vector $W$, which has a length of $k$, $P(w_i \text{ is at position } j) = \frac{1}{k}$, where $P$ represents the probability, $w_i$ denotes the $i$-th word, $j$ indicates the position within the input vector, and $k$ is the total length of the input vector $W$.

This homogeneous formation of the input contributes to the merging and mixing of information, making it challenging to track and interpret the output. Despite the transparent nature of the TM structure and its ability to clearly explain the reasoning behind any output, the process of shuffling input target words confuses the model, compromising transparency and interpretability. Another issue with that training method is that the larger the input vector, the longer the training time.

In practical NLP applications, the embedding method in Bhattarai et al. (2024) is not scalable or reusable because the output cannot be effectively leveraged. For example, in LLMs such as GPT-3, one of the initial steps after receiving the input vector is to generate embeddings for each word. These embeddings include context information extracted from documents that the model was trained on, and are stored as dense vector representations in a high-dimensional space. Unlike GPT-3, which retains these embeddings for efficient reuse across tasks, the TM-AE model Bhattarai et al. (2024) generates embeddings specific to the input target words and can not be used for different input combinations. In this work, the aforementioned problems were identified, and a TM-AE was employed in two phases, each with a distinct structure to construct the input $X$ and then trained

using the CoTM structure to represent embeddings for a vector of target words. Fig. 3 illustrates the two phases used for this application.

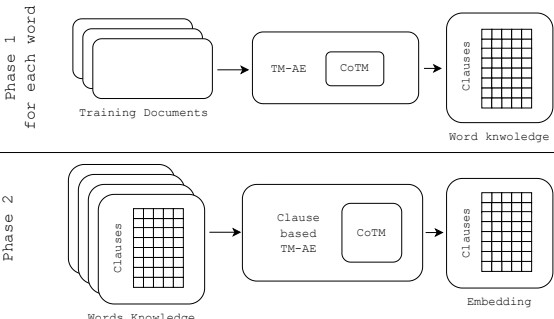

Figure 3: Proposed two-phase architecture based on TM-AE for embedding: Phase 1 involves training documents per word and for all the vocabulary, and Phase 2 focuses on clause-based embedding with word knowledge.

### 3.3 PHASE 1

The first phase involved the original TM-AE architecture, with the key difference being that the model was trained with the target words individually, meaning each training instance involved a single word. This approach offered several advantages:

1. Extracting the knowledge in a pure form for the target word, thereby maintaining transparency in interpreting any training outputs.

2. Enabling the storage of training results for use in other applications, similar to models like Word2Vec. In this research, the stored results were used as a dataset for the second phase.

3. Facilitating the application of real-world scenarios that can be updated in the future. If it becomes necessary to update the context information for a word in the vocabulary, it is straightforward to retrain only that specific word.

The training results from this phase consist of conjunctive clauses that describe the target word using specific literals in propositional logic. For example, training on the word "car" resulted in 1600 clauses, such as "driver AND road." The process of forming the input $X$ in this manner is characterized by the randomness inherent in selecting documents that either support the target word (resulting in a training outcome of 1) or do not support it (resulting in a training outcome of 0). This randomness facilitates the extraction of different forms of contextual information for the target word from both supporting and non-supporting documents without the interference of other words, as was the case in the previous work.

To formalize this, for each word $word_k$, the number of documents selected is the minimum between a window size $a$ and $|D_{word_k}|$, i.e., the actual number of available documents. Therefore, the number of documents picked for $word_k$ is given by $\min(a, |D_{word_k}|)$. The total number of words extracted from the picked documents depends on the number of documents selected and the words contained in each document. Let $d_i$ represent a document, $\text{Words}(d_i)$ be the number of words in document $d_i$, and $D_{\text{picked}}$ be the set of documents selected for $word_k$. The total number of words extracted for $word_k$ is calculated as $\sum_{d_i \in D_{\text{picked}}} \text{Words}(d_i)$, which sums up the number of words from each document in the selected set to build $X$.

---

**Algorithm 1** Building $X$ in Phase 2 using the knowledge from Phase 1

---

**Require:** Phase 1 knowledge as clauses $C$, vocabulary $V$, target words $W$ (with $k$ elements), number of examples $r$, target values $q \in \{0,1\}$, window size $a$, subset clauses $S \in C$, documents of literals $D_l$

1: **for** each epoch **do**
2:     **for** $i = 1$ to $r$ **do**
3:         Shuffle target words $W$, and randomly select $q \in \{0,1\}$
4:         **for** each word $word_k$ in $W$ **do**
5:             Initialize empty set $X$. Load knowledge data for target word $word_k = C_{word_k}$
6:             **if** $q = 1$ **then**
7:                 Filter clauses: $C^+_{word_k} = \{$clauses with positive weights$\}$
8:             **else**
9:                 Filter clauses: $C^-_{word_k} = \{$clauses with negative weights$\}$
10:             **end if**
11:             Sample subset $S_c$ from filtered clauses $C_{word_k}$ of size $a$
12:             **for** each clause $C_j \in S_c$ **do**
13:                 **for** each literal $l_{ij} \in C_j$ **do**
14:                     Append $l_{ij}$ to $D_l$, and load knowledge data for literal $l_{ij} = C_{l_{ij}}$
15:                     **if** $q = 1$ **then**
16:                         Filter clauses: $C^+_{l_{ij}} = \{$clauses with positive weights$\}$
17:                     **else**
18:                         Filter clauses: $C^-_{l_{ij}} = \{$clauses with negative weights$\}$
19:                     **end if**
20:                 Sample subset $S_l$ from filtered clauses $C_{l_{ij}}$ of size $a$
21:                 **for** each clause $C_k \in S_l$ **do**
22:                     **for** each literal $l_{ik} \in C_k$ **do**
23:                         Append $l_{ik}$ to $D_l$
24:                     **end for**
25:                 **end for**
26:                 **end for**
27:             **end for**
28:             Activate literals in $X$ by taking $D_l$ literals, and update CoTM
29:         **end for**
30:     **end for**
31: **end for**

---

### 3.4 PHASE 2

The purpose of training in Phase 2 is to find embedding for a set of input target words. In the second phase, we utilize the knowledge and vocabulary index obtained from the first phase to generate the input $X$ and then do the training. Algorithm 1 describes the construction method. The knowledge for each target word consists of a set of clauses $C$. These clauses are divided into positive weight clauses that vote for the target word and negative weight clauses that vote against it.

Suppose we have an input vector $Words = [word_1, word_2, word_3]$. In each epoch, and with each example of $r$, the target words in $Words$ are randomly arranged to obtain a high level of homogeneity in mixing the context information. For example, if the first example has the order $Words = [word_1, word_2, word_3]$, the second example can have the order $Words = [word_2, word_3, word_1]$. This ensures that each target word has an equal probability of appearing in any position in the vector.

For each word in the input vector, the knowledge associated with that word is retrieved $C_{word_k}$. Using another random process, training is performed on a target with an output of 1 or 0, retrieving positive or negative clauses, respectively. Based on the user-set window size $a$, a subset of these clauses is selected $S_c$, and all the literals within those clauses are extracted. These literals act as documents in the first phase, where all the clauses associated with them are fetched. Finally, the literals collected from these clauses, after taking a subset $S_l$ of size $a$, are collected to construct $X$.

The second phase can be viewed as a clause-to-clause encoding process, where the input and output are clauses. This hierarchical tree of knowledge fetching is interpretable and transparent, which

maintains the reasoning behind the output. It is important to note that in this phase, there is no need to compute the negation of features, as the results from Phase 1 already encompass these literals. This eliminates redundant calculations and ensures that both positive and negative representations of the features are inherently captured within the clause construction process.

The main difference between the first phase and the second phase lies in the construction of the input $X$ and the source of information. The output of the second phase is an embedding of the input target words based on the knowledge collected in the first phase. This embedding is then utilized to measure similarity with a dataset prepared by humans, as will be further demonstrated in the subsequent section. In future applications, the first phase can be utilized for other operations depending on the specific application, as demonstrated in our experimental results section, particularly in data augmentation tasks for sentiment analysis.

To demonstrate the output of Phase Two in terms of transparency and interpretability, we can use the words "drive" and "road" as an example. The output of the first phase for the word "drive" includes clauses such as "vehicle AND license" or "road AND safe," while the knowledge associated with the word "road" includes clauses like "vehicle AND smooth" and "driver AND traffic." After embedding and training in the second phase, the resulting output for the word "drive" might include clauses such as "vehicle AND (license OR smooth)" or "(road AND safe) OR (driver AND traffic)." The interpretability of this output is evident in the ability to trace the reasoning behind the inclusion of specific words in the second phase. These words emerge from the integration of clauses derived from the knowledge of common words, which themselves are interpretable and transparent, providing clarity regarding their presence in the final output.

## 4 EMPIRICAL RESULTS

To evaluate our proposed approach, we conducted two types of assessments. First, we used human-annotated similarity evaluation datasets to compare the similarity scores generated by the Phase 2 embeddings. Second, we applied the Phase 1 knowledge results to a sentiment analysis task for data augmentation, measuring the quality of our approach in comparison to DL models.

### 4.1 SIMILARITY EVALUATION

We assessed the performance of Phase 2 embeddings through a series of benchmark experiments using widely recognized human-annotated datasets, including RG65, MTURK287 (MT287), MTURK717 (MT717), and WS-353 (WS353). These datasets contain word pairs with assigned similarity scores, covering a broad range of semantic relationships. Our experiments aimed to compare the similarity of Phase 2 embedding with these datasets. We used various similarity measures, such as Cosine similarity, Spearman, and Kendall, which are statistical measures used to compare the similarity or correlation between different datasets or vectors.

Table 1: Comparison of Spearman (S), Kendall (K), and Cosine (C) similarity measures.

| Dataset | Word2Vec | | | Fast-Text | | | TM-AE | | | Two-Phase TM-AE | | |
|---|---|---|---|---|---|---|---|---|---|---|---|---|
| | S | K | C | S | K | C | S | K | C | S | K | C |
| WS353 | 0.58 | 0.41 | 0.91 | 0.46 | 0.31 | 0.77 | 0.38 | 0.24 | 0.86 | 0.41 | 0.29 | 0.90 |
| MT287 | 0.55 | 0.38 | 0.86 | 0.58 | 0.41 | 0.68 | 0.53 | 0.36 | 0.88 | 0.40 | 0.28 | 0.97 |
| MT717 | 0.47 | 0.32 | 0.87 | 0.41 | 0.28 | 0.63 | 0.46 | 0.32 | 0.88 | 0.34 | 0.23 | 0.95 |
| RG65 | 0.51 | 0.34 | 0.84 | 0.43 | 0.30 | 0.68 | 0.63 | 0.45 | 0.87 | 0.50 | 0.42 | 0.91 |
| Avg. | 0.52 | 0.36 | 0.87 | 0.47 | 0.32 | 0.69 | 0.50 | 0.34 | 0.87 | 0.41 | 0.30 | 0.93 |

Table 1 shows the evaluation results compared to related algorithms. Despite the potential for scaling and transparency in extracting knowledge for each word in the first phase, our results indicate that the old method is generally more accurate when evaluated using pre-prepared datasets. There are two main reasons for this discrepancy. Firstly, the old method extracts common context information between words by training the TM to fuse and blend all the information in the input samples and find the commonality with respect to the input vector. The training focuses on extracting context

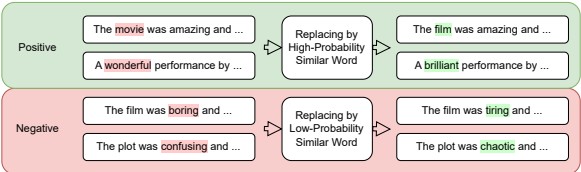

Figure 4: Examples of sentence augmentation using word embeddings for sentiment analysis.

information for the input vector as a whole rather than general knowledge for each word in the input vector. In contrast, the first phase of our proposed approach trains the TM in an unsupervised way with the goal of extracting context information for the target word. This may bias the output based on the type of data available in the database and the size of the database. Secondly, the size of the database is crucial. In our proposed approach, the first phase involves training with a specific hyperparameter to extract knowledge that determines finding the embedding for the input vector in the second phase. Even if we were to expand and increase the values of the training for the second phase, it would still be limited by the knowledge extracted from the first phase.

It is noteworthy that the model was trained over a period of six months on a DGX H100 server to achieve the reported results. The training was conducted using the One Billion Word dataset, which consists of a vocabulary size of 40k words. Each word in the vocabulary was trained with the following configuration: 2,000 examples per epoch, a window size of 25, 1600 clauses, a threshold $T$ of 3,200, a specificity parameter $s$ set to 5.0, and across 25 epochs. While extracting context data in the first phase is crucial, it can be a time-consuming process as training must be conducted on all words in the database vocabulary. However, this process only needs to be done once. Furthermore, in future updates, updating a single word may not require new training for the entire first phase, but only for the targeted words. For example, let's suppose we have a database related to healthcare where the word "cancer" is a target word. If a new study comes out that shows a strong correlation between a certain food and cancer, the knowledge associated with the word "food" may need to be updated to reflect this new information. This would only require updating the "food" knowledge rather than retraining the entire first phase.

Notably, in terms of Cosine similarity, the two-phase TM-AE achieved 0.91 on the RG65 dataset, surpassing all Word2Vec (0.84), FastText (0.69) and the TM-AE (0.87). This indicates that the two-phase approach effectively captures semantic relationships when similarity is assessed through vector alignment. However, its performance in Spearman and Kendall correlations is comparatively weaker. For instance, in the WS-353 dataset, it scores 0.41 and 0.29, respectively, which is lower than Word2Vec's scores of 0.58 and 0.41. This suggests that while the two-phase model produces closely aligned embeddings, it may not accurately reflect the rank order of word pairs as judged by humans. Furthermore, the two-phase model exhibits variable performance across datasets. For example, it achieves a high Cosine similarity of 0.97 in the MTURK287 dataset but experiences a decline in Spearman correlation to 0.40. Overall, while the two-phase TM-AE shows promise in generating aligned embeddings, further improvements should focus on optimizing the knowledge extraction phase to ensure better alignment with human semantic evaluations across diverse datasets.

## 4.2 SENTIMENT ANALYSIS

In NLP, embeddings can be leveraged for sentiment analysis using data augmentation by assessing their quality on unseen data and evaluating the impact of changes in the trained dataset. This method has been applied with various counterparts of popular DL models to benchmark the effectiveness of the proposed embeddings. Given that such applications typically rely on classification models, this work will mark the first instance of incorporating multiple TM structures within a single application. Fig. 4 illustrates the methodology for document augmentation using word embeddings in sentiment analysis. In positive reviews, words were substituted with highly similar, high-probability words from the embedding (e.g., "movie" replaced with "film"). Conversely, in negative reviews, words were replaced with less similar, low-probability alternatives (e.g., "confusing" replaced with

"chaotic"). These nuanced modifications demonstrate how embedding-based word replacements influence both positive and negative sentiment.

Table 2 presents our experiments' accuracy results using the IMDB dataset, which consists of 25K training samples and an additional 25K samples for evaluation. The data augmentation experiments were performed using various embedding models, including GloVe, Word2Vec, FastText, ELMo, BERT, and Two-phase TM-AE. The performance was evaluated using several classifiers: Logistic Regression, Naive Bayes, Random Forest, Support Vector Machine, Multi-Layer Perceptron, and Tsetlin Machine.

For the TM model, the embedding was generated during the first phase by training on a vocabulary of 20K words of the IMDB dataset. In this phase, the hyperparameters were set as follows: 2,000 examples, a window size of 25, 800 clauses, a threshold $T$ of 1,600, a specificity parameter $s$ of 5.0, and 25 epochs. In the classification phase using the TM classifier, the settings were configured to 1,000 clauses, a threshold $T$ of 8,000, a specificity parameter $s$ of 2.0, and 10 epochs. These configurations enabled a robust comparison of embedding models across different classifiers, showcasing the effectiveness of the TM-AE embedding and classifier architecture.

Table 2: Comparison of accuracy using different embedding sources and classifier: LR (Logistic Regression), NB (Naive Bayes), RF (Random Forest), SVM (Support Vector Machine), MLP (Multi-layer Perceptron), and TM (Tsetlin Machine)

| Embedding Source | Classifiers (accuracy) | | | | | |
|---|---|---|---|---|---|---|
| | LR | NB | RF | SVM | MLP | TM |
| GloVe | 0.72 | 0.82 | 0.68 | 0.70 | 0.72 | 0.60 |
| Word2Vec | 0.80 | 0.82 | 0.76 | 0.80 | 0.83 | 0.73 |
| FastText | 0.75 | 0.83 | 0.75 | 0.74 | 0.75 | 0.74 |
| Two-phase TM-AE | 0.80 | 0.80 | 0.79 | 0.79 | 0.81 | 0.78 |
| BERT | 0.83 | 0.82 | 0.82 | 0.80 | 0.80 | 0.82 |
| ELMo | 0.83 | 0.83 | 0.84 | 0.81 | 0.84 | 0.83 |

The results in Table 2 provide a comprehensive comparison of the performance of various embedding models across different classifiers. Notably, the two-phase TM-AE model shows competitive accuracy, especially with the TM classifier, achieving an accuracy of 0.78, surpassing GloVe (0.60) and FastText (0.74). However, The two-phase TM-AE falls behind models such as ELMo and BERT. ELMo's dynamic contextualized embeddings adapt to each word's context in a sentence, unlike the static embeddings used in the two-phase TM-AE approach. ELMo's approach consistently selects similar or dissimilar words from a limited range of options within individual documents, as its embeddings are generated per document rather than across the entire dataset. This localized embedding strategy impacts the augmentation of the training set, subsequently influencing the classification results. As for BERT, its superior performance is attributed to its bidirectional nature, which allows it to capture context from both directions in a sentence, leading to richer contextualized embeddings. Additionally, BERT's transformer-based architecture enables it to model long-range dependencies more effectively, further enhancing classification accuracy.

The two-phase TM-AE shows promising results, especially with TM classifier, offering a propositional-logic-based transparent end-to-end architecture. However, there is room for improvement when compared to more established models like BERT and ELMo, suggesting further refinement of the embedding and classification processes.

## 5 CONCLUSION

We proposed a novel two-phase approach to word embedding based on the TM, designed to improve scalability for NLP tasks. By leveraging CoTM, the model captured contextual knowledge of words in a structured logical form. Our evaluation demonstrated competitive performance in similarity benchmarks and sentiment analysis where embedding quality was validated through data augmentation. Together with TM classifier, we introduced, for the first time, an end-to-end scalable, transparent, and propositional-logic-based approach, paving the way for its use in a variety of NLP applications.

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

# 6 APPENDIX

## 6.1 TM-AE ALGORITHM

The following algorithm 2 outlines the process used in TM-AE for constructing the input $X$ from the supporting documents in Bhattarai et al. (2024) framework.

---

**Algorithm 2** Building $X$ from supporting documents in TM-AE by Bhattarai et al. (2024)

---

**Require:** Documents $D$, vocabulary $V$, target words $W$ (with $k$ elements), number of examples $r$,
    target values $q \in \{0, 1\}$, window size $a$

1: **for** each epoch **do**
2:    **for** $i = 1$ to $r$ **do**
3:       Shuffle target words $W$, and randomly select $q \in \{0, 1\}$
4:       **for** each word $word_k$ in $W$ **do**
5:          Initialize empty set $X$
6:          **if** $q = 1$ **then**
7:             Select $a$ random documents from $D$ that contain $word_k$
8:          **else**
9:             Select $a$ random documents from $D$ that do **not** contain $word_k$
10:         **end if**
11:         Combine the selected documents. Extract all words from the combined documents
12:         Activate literals in $X$ by including the extracted words and their negations.
13:         Update CoTM
14:       **end for**
15:    **end for**
16: **end for**

---

## 6.2 FUTURE WORK

In this work, we have presented an optimal approach for initiating the development of a practical embedding that can be leveraged in various applications. The application of sentiment analysis through data augmentation (4.2) represents the first instance in which the embedding method proposed in the first phase was utilized. Looking ahead, our future efforts will focus on enhancing the efficiency of the embedding, improving model performance, and addressing the current limitations associated with the slow implementation.

We also recognize that the embedding requires further expansion to better capture contextual information. Achieving this will involve revising the manner in which the TM constructs clauses, as the current embedding heavily relies on this process. Such enhancement will allow for improvements in the second phase, specifically in calculating word similarity, as discussed in this study. Additionally, ensuring a deeper and more structured first phase will significantly contribute to applications tailored to domain-specific tasks. For example, in medical NLP, the ability to incorporate embeddings for newly introduced terms without the need to retrain the entire model would enhance scalability and adaptability.

Furthermore, we anticipate that our approach will have applications in Knowledge Graph Construction, where the interpretability of propositional clauses would be particularly valuable. Such interpretability makes the resulting embeddings well-suited for tasks such as semantic search, question-answering, and reasoning.

