# OpenReview forum: "Scalable Multi-phase Word Embedding Using Conjunctive Propositional Clauses"
_ICLR.cc/2025/Conference — Submitted to ICLR 2025_

### Official Review · Reviewer_cBmB · 2024-10-28

**Soundness:** 2
**Presentation:** 1
**Contribution:** 2
**Rating:** 5
**Confidence:** 2

**Summary:**

The paper introduces a scalable two-phase word embedding approach using TM and conjunctive propositional clauses. Phase 1 captures knowledge for individual words, while Phase 2 constructs embeddings for word sequences. Experiments show that the model achieves competitive cosine similarity on benchmark datasets but lags in Spearman correlation. Additionally, the model performs moderately well in sentiment analysis compared to BERT and ELMo. This approach demonstrates TM’s potential for interpretable NLP applications, though further refinement is needed.

**Strengths:**

- The two-phase approach to word embedding with TM is novel, especially using conjunctive propositional clauses, which adds interpretability.
- The use of Tsetlin Machines enables a more transparent and interpretable embedding process compared to deep learning models.
- Practical Application in Sentiment Analysis: The embeddings show utility in real-world tasks, as demonstrated through sentiment analysis with data augmentation.

**Weaknesses:**

- The results don’t look good; the model performs poorly on Spearman and Kendall correlations, which suggests it struggles to capture the ranking of word pairs.
- I guess Algorithm 1 doesn’t seem to be a direct contribution of this paper, so it might be better left out of the main text or put into Appendix.
- The paper could use more polish. For example, Tables 1 and 2 are missing top and bottom lines, which affects readability.
- The evaluation is pretty narrow, focusing mostly on basic similarity metrics and sentiment analysis, without exploring a broader range of NLP tasks that would really show the model’s robustness.
- For Table 2, why the socres for all models looks quite similiar? Also, I didn't see the TM outperform the other models. It even lag behind a lot for some embeddings. It may happens very randomly but it becomes hard to convince the advantages of TM.

**Questions:**

See the Weaknesses

---

> ### Author Response · Authors · 2024-11-25
>
> We acknowledge the weaknesses highlighted by the reviewer, which provide critical insights into our work, and we sincerely thank them for their thoughtful feedback and effort. Below, we address these points in detail:
>
> 1. An additional experiment was included in the similarity evaluation table for the FastText model, which is considered an enhanced version of Word2Vec. Interestingly, in this specific application, FastText yielded weaker results. This observation suggests that the classification adopted, while crucial for performance evaluation, relies heavily on human-constructed databases. These datasets, though widely accepted, may lack objectivity for this particular application and are generally subject to human biases. However, the primary focus of our work lies in the performance differences between the new model and the previous one by Bimal. These differences are significant and merit detailed analysis, which has been provided in **Section 4.1, paragraph two**.
>
> 2. The algorithm has been relocated to the appendix for better organization and accessibility.
>
> 3. The tables in the manuscript have been updated and improved.
>
> 4. During the preparation of our responses, we attempted to incorporate an additional application—document clustering. However, after initiating the application and observing preliminary results, we realized it would be infeasible to complete this work before the deadline due to time constraints. Nonetheless, we appreciate the reviewer’s suggestion and recognize the importance of exploring this application in the future.
>
> 5. The experiment related to the two-phase TM was repeated with an increased vocabulary size, expanded from 10K to 20K. The earlier vocabulary was limited and insufficient, leading to weaker results and an unfair comparison with prior models. With the expanded vocabulary, the results showed significant improvement. Regarding the convergence of results between models, this could be attributed to the nature of the application and the advanced state of the models employed, which are already optimized to perform competitively. The source code for these experiments is attached in the Supplementary Material. Please note that all source codes will be openly available online once the anonymous review period concludes.

---

### Official Review · Reviewer_miWg · 2024-11-03

**Soundness:** 2
**Presentation:** 2
**Contribution:** 2
**Rating:** 3
**Confidence:** 4

**Summary:**

This paper presents a scalable two-phase word embedding method using the Tsetlin Machine (TM) framework for NLP tasks. Phase 1 generates interpretable word embeddings by learning contextual knowledge for each target word, capturing it in logical clauses. In Phase 2, these embeddings are used to create representations for word sequences, supporting applications like sentiment analysis. Evaluations show competitive performance, particularly in sentiment analysis and similarity assessments on human-annotated datasets. The approach provides a transparent, reusable alternative to traditional embeddings.

**Strengths:**

A novel tsetlin machine based incorporating two-phase training to discover contextual embeddings of input sequences.

**Weaknesses:**

- The main motivation is not clearly expressed; the introduction does not outline the differences and advantages over previous methods, only presenting an overview of prior work. The CoTM method is also not original, thus the interpretability mentioned at the beginning is also not a contribution of this paper.

- The introduction mentions the issue of long training time, but the experiments section does not seem to analyze the efficiency issue.

- The experiments are conducted only on a sentiment analysis dataset for a text classification task, which is of limited persuasiveness.

**Questions:**

I have a question regarding the Tsetlin machine-based word embedding approach. This method does not seem to account for word order, relying more on co-occurrence features to represent semantics. As a result, it may be less effective at capturing dependency, structure, and other linguistic characteristics. This could be why it performs reasonably well only on tasks like sentiment classification that do not require deep structural semantics. What is the authors' view on this issue?

---

> ### Author Response · Authors · 2024-11-25
>
> We sincerely appreciate the reviewer’s insightful comments, which have significantly contributed to refining our work. Below, we provide detailed clarifications addressing the highlighted weaknesses and questions:
>
> ### **Weaknesses**
>
> 1. Firstly, in the introduction, we specifically emphasized a primary limitation of the previous method of embedding generation using TM. The earlier approach is computationally intensive and produces embeddings with limited reusability, impeding integration with larger systems, such as large language models, where embeddings need to be adaptable for use across various stages and applications. This constraint is a key barrier to scalability, underscoring the need for a more versatile embedding framework.
>
>    Additionally, in paragraph 3 of the introduction, we described an experiment designed to illustrate the limitations of the prior method in the context of embedding scalability. Unlike traditional deep learning techniques, the previous approach does not easily extend to applications that incorporate multiple layers, which informed our primary research objective: to develop a scalable TM-based embedding method. Our contributions outline how this two-phase approach facilitates the construction of scalable embeddings (first contribution). We demonstrate this in Section 4.1 with multi-word vector evaluation (second contribution), and further validate it through data augmentation applications in Section 4.2 (third contribution).
>
> 2. This question is indeed critical, as we encountered challenges in comparing the embedding collections across the entire dataset vocabulary used in the new approach versus those in the previous approach. The previous method lacks comprehensive extraction of contextual information for the vocabulary included in training; rather, it focuses on specific elements directly related to the training task, without maintaining the primary advantage of transparency inherent in Tsetlin Machines (as noted in Section 3.2).
>
>    Therefore, the extended execution time of the previous method was highlighted to illustrate that, despite its lengthy runtime, the generated embeddings are limited in their applicability to other experiments, even with minor modifications such as altering a single word involved in training. This limitation underscores the lack of scalability in the previous model. In contrast, the proposed model necessitates comprehensive training across all vocabulary in the dataset, which indeed requires a longer initial training period, but this process is only performed once; the resulting embeddings can then be directly applied across various applications.
>
> 3. The research involved two primary databases. The first is the One Billion Words corpus, which was utilized to assess pairwise similarity (as described in Section 4.1) based on four human-annotated similarity evaluation datasets. The second database is the IMDB dataset, focused on sentiment analysis (Section 4.2), and was applied in data augmentation tasks. In the updated version, an additional comparison has been included in the first table, and the second table has been updated with higher-quality embeddings derived from the IMDB dataset.
>
> ### **Questions**
>
> In preparing the input \( X \) at each phase of the model, data is converted into sparse matrices, which streamlines the processing and execution of logical operations central to the TM framework. During this conversion, word frequency within each document was disregarded at both phases. This approach implies that the model does not factor in word occurrence frequency; instead, it leverages the principles of Tsetlin Automata to select features (words) that are representative of the trained content.
>
> Although this aspect may have been implicitly addressed within the foundational TM structure, the significance of disregarding word frequency was unfortunately not explicitly stated in either the previous or current methodologies. We appreciate the reviewer’s insight on this point and confirm that this clarification will be explicitly added to **Section 3.1**:
> > “It is important to note that the construction of \( X \) is independent of the frequency with which a word appears in a given document; its mere presence, even if mentioned only once, is sufficient.”
>
> Regarding the embedding’s structural depth and semantic richness, we consider the generated embeddings to be sufficiently deep and descriptive for comparison with prior methods, aligning closely with the examples provided in the previous approach. However, we acknowledge that the embedding may not capture a densely interconnected representation of word relationships, a limitation which has been noted in the appendix as a direction for future research.

---

### Official Review · Reviewer_aGqt · 2024-11-04

**Soundness:** 2
**Presentation:** 2
**Contribution:** 2
**Rating:** 5
**Confidence:** 2

**Summary:**

This paper presents a novel two-phase method for generating word embeddings using the Tsetlin Machine (TM) and Coalesced Tsetlin Machine (CoTM) structures. The authors describe how the CoTM automaton process and the TM autoencoder (TM-AE) operate, introducing a two-phase training mechanism for the TM-AE. The proposed approach's effectiveness is evaluated on semantic similarity and sentiment classification tasks, comparing the performance of the two-phase TM-AE embeddings with other word embedding frameworks, including GloVe, Word2Vec, FastText, BERT, and ELMo. Results indicate that the two-phase TM-AE embeddings outperform GloVe, Word2Vec, and FastText on specific tasks and metrics. Additionally, the authors highlight that this approach represents the first scalable, transparent, end-to-end framework for generating word embeddings based on propositional logic through a TM-based method.

**Strengths:**

This paper has several notable strengths:
1. It introduces a relatively novel approach by employing the Tsetlin Machine automaton to generate word embeddings, distinguishing it from more conventional methods.
2. The paper provides a thorough explanation of the CoTM architecture, TM automaton, TM-AE training process, and the unique two-phase TM-AE training schema.
3. The authors highlight a computational efficiency feature: by skipping the first phase of TM-AE when updating a single word, one can reduce the computational load.
4. Multiple human-annotated datasets, including RG65, MTURK287, MTURK717, and WS-353, are used to rigorously evaluate the quality of the generated word embeddings.
5. The proposed two-phase TM-AE embeddings are comprehensively compared to established models such as GloVe, Word2Vec, FastText, BERT, and ELMo, offering insight into its relative strengths and weaknesses.

**Weaknesses:**

The paper has several areas where it could be improved:
1. Although the model reportedly required 6 months of training on a DGX H100 machine, the paper lacks an analysis of the computational time and space complexity of the proposed method, particularly in comparison to other approaches like GloVe, Word2Vec, and FastText.
2. While the paper emphasizes the scalability and transparency of the two-phase TM-AE approach, further explanation and analysis of these properties would strengthen the argument for its advantages. For example, present empirical scalability results with increasing vocabulary sizes, or provide examples of how the logical structure of embeddings can be interpreted.
3. The quality of the two-phase TM-AE embeddings could be better demonstrated by including additional evaluation tasks beyond semantic similarity, such as analogy, categorization, and outlier detection.
4. The hyper-parameters used in the sentiment classification MLP model and TM classifier are not documented, and there is no discussion of the hyper-parameter fine-tuning process for the TM classifier.
5. In Section 4.1, the comparison of TM-AE and two-phase TM-AE on similarity tasks shows that TM-AE performs better on two out of three metrics. This result raises questions about the benefit of the two-phase approach for this task. To address this, you may provide a more in-depth analysis of why the two-phase approach underperforms on these metrics, or what potential trade-offs or benefits it might offer despite these results.
6. In Section 4.2, the paper omits the performance of the TM-AE model, which would provide valuable context for assessing the gains offered by the two-phase TM-AE.

**Questions:**

In addition to addressing the items raised in the Weaknesses section, the authors could consider the following suggestions to further enhance the paper:
1. Include a more detailed comparison between TM-AE and two-phase TM-AE embeddings to clarify the specific benefits and limitations of each approach.
2. Evaluate the quality of two-phase TM-AE embeddings on additional word embedding tasks such as analogy, categorization, and outlier detection.
3. Provide a few examples of two-phase TM-AE embeddings to illustrate their characteristics more concretely.
4. Demonstrate the potential of two-phase TM-AE embeddings in more complex NLP tasks by incorporating them into Transformers or other state-of-the-art NLP or large language models, showcasing applications beyond classification tasks.
5. Include a discussion of possible future research directions to give readers insight into how this work could be further developed.
6. Provide the code for two-phase TM-AE training and evaluation to facilitate reproducibility and further research.

---

> ### Author Response · Authors · 2024-11-25
>
> We sincerely thank the reviewers for their efforts in helping us improve the quality of our research. The points raised also address several key weaknesses, and we provide our responses as follows:
>
> ### **Weaknesses**
>
> 1. Referring to the lack of analysis of the computational complexity (time and space) of the proposed method, we acknowledge some challenges encountered with our approach:
>    - The source code includes several fundamental sequential operations, which hinder the possibility of parallel execution to leverage faster processing. Despite utilizing six GPU cores, the performance gain was minimal. We recognize this limitation and aim to address it in future work.
>    - In the proposed model, hypervector concepts were employed in a basic manner. During the course of our work, we discovered that a research team had developed more advanced techniques to enhance the performance and storage efficiency of Tsetlin Machines using hypervectors. Incorporating such techniques in future iterations of our model could lead to substantial improvements (https://arxiv.org/abs/2406.02648).
>    - Profiling operations revealed that a significant portion of the execution time is consumed by I/O logic used to store files in a pickled manner. By adopting more efficient data storage methods and reducing I/O operations, we anticipate performance improvements. This enhancement is also part of our future development plans.
>
> 2. During our preparation for this response, we attempted to include an additional application: document clustering. However, upon initiating this task and observing the preliminary results, we realized that completing this work before the deadline was unfeasible due to time constraints. Nonetheless, we appreciate the reviewer’s suggestion and acknowledge its importance.
>
> 3. Concerning the weakness related to the hyper-parameters of the TM Classifier, we confirm that this information is included in **Part 4.2, paragraph 3**. However, the hyper-parameters related to MLP were not discussed, as the settings were default across all experiments (with `random_state=42`). This can be verified in the accompanying source code.
>
> 4. Please refer to the response provided for **Question 1** below.
>
> 5. We have re-conducted the experiments, increasing the vocabulary size from 10k to 20k. Previously, the smaller vocabulary resulted in weak embeddings and an unfair comparison. With the expanded vocabulary, there was a significant improvement in the results.
>
> 6. Please see the responses to the questions points below.
>
> ### **Questions**
>
> 1. Since the hypothesis in our research revolves around the previous method of building embeddings using TM principles, the differences between the two approaches were discussed in depth throughout the manuscript. Below, we provide a concise summary of these distinctions:
>    - **Introduction (Third paragraph onward):**
>      We conducted an experiment focused on the primary research question: whether the long training time in the previous method can be mitigated to allow reusability. Based on this, we outlined three contributions of the research:
>        1. Solving the problem of scalability and enabling the reuse of training results.
>        2. Constructing embeddings for input word sequences to facilitate comparison with the previous method.
>        3. Highlighting one potential application of our proposed method.
>    - **Section 3.2 (Final paragraph):**
>      We identified another issue in the previous approach: the fusion of information and loss of TM's transparency, which is addressed in our method.
>    - **Section 3.3:**
>      The advantages of transitioning the old method into the first phase of the new model were highlighted in three points. An example using the word “car” demonstrated our method’s ability to retain interpretability after embedding, thus preserving the transparency of TM compared to the previous approach.
>    - **Section 4.1 (Second paragraph):**
>      While discussing the results, we provided a rationale for why the previous method outperformed the new method in certain similarity measures.
>
> 2. Please refer to the response provided under **Weakness 3** above.
>
> 3. Examples have been added in **Section 4.2** to clarify the embedding results from the two phases.
>
> 4. Additional points on expected future work have been included in the appendix.
>
> 5. Please refer to the response provided for **Weakness 1** above.
>
> 6. The source code will be publicly available online. However, due to conference anonymity requirements, it was not explicitly mentioned in the manuscript. To address this, we provide the full source code in the Supplementary Material for your review.

---

> ### Comment · Reviewer_aGqt · 2024-12-01
>
> I would like to thank the authors for their response.
>
> Among the suggestions I provided, only 2 items are replied with updated discussions in the paper and comments, including comparison between TM-AE and two-phase TM-AE, and future research directions. While other items are not updated with substantial changes, especially, the following are still missing:
> 1. Computation complexity of two-phase TM-AE
> 2. Additional evaluation tasks beyond semantic similarity, such as analogy, categorization, and outlier detection
> 3. Although the authors mentioned they have updated Table 1 after increasing the vocabulary size from 10k to 20k, I did not numbers within Table 1 being updated
> 4. Experiment results of TM-AE within Table 2
> 5. Documentation of hyper-parameter tuning of MLP within section 4.2
>
> The above updates provided by the authors are not sufficient for me to raise my rating from 3 to 5.

---

> > ### Author Response · Authors · 2024-12-03
> >
> > We extend our gratitude to the reviewer for their valuable feedback and confirm our commitment to addressing the majority of the points raised as follows:
> >
> > 1. **Computational Complexity**
> > The following text will be included in Section 3.2 to address the computational complexity:
> >
> > > When comparing the computational complexity of the previous and proposed methods, the following observations can be made:
> > > a. Both methods leverage the TM for training and deriving output via propositional logic. Both have demonstrated efficiency, speed, and manageable computational complexity in prior research (Maheshwari et al. (2023)).
> > > b. The first phase of the proposed method aligns with the previous method in terms of computational complexity and execution time, as they share identical source code. The key difference lies in the training objective: the new method extracts embeddings for individual input words, whereas the previous method extracts embeddings for a set of words.
> > > c. The second phase introduces a novel solution to improve upon the embedding approach in the previous method. Despite differences in the input X preparation—where the previous method uses a dataset of documents and the new method employs clauses—both share similarities in how the model processes the input. Key challenges in both methods include:
> > > i. The presence of numerous sequential operations in the source code that hinder parallelization and fast execution.
> > > ii. The use of hypervector concepts in the proposed model in a relatively simple form. Future work aims to incorporate advanced methods, such as those discussed in Halenka et al. (2024), to enhance training efficiency.
> > > iii. Profiling results of this work indicate that a significant portion of execution time is consumed by I/O logic for file storage in pickle format. A temporary solution involved caching files in RAM, but the adoption of more efficient data storage techniques and a reduction in I/O operations are planned to improve performance.
> >
> > 2. **Application Addition**
> > Regrettably, we are unable to include the suggested application at this time due to time constraints.
> >
> > 3. **Tables and FastText Results**
> > We acknowledge the oversight in the response regarding the first table. In this table, a column has been added to include similarity test results using FastText. This experiment underscores the distinction between the new model and the previous model by Bhattarai et al. (2024). Explanations for these results have been elaborated upon in Part 4.1, Paragraph Two.
> > The text has also been updated to include the following:
> >
> > > When comparing similarity results between Word2Vec and FastText—which is expected to be an improved version of Word2Vec (Bojanowski et al. (2017))—it was observed that FastText produced weaker results in this specific application. This suggests that while the adopted classification is critical for evaluating our model's performance, it heavily relies on human-created databases. These databases, while generally weighted to measure similarity between words, may not be entirely objective for this application.
> >
> > 4. **Representation in Tables**
> > Due to the nature of the TM-AE model design, it is unsuitable for this application, as it does not generate vectors or context information for individual words. This limitation is one of the primary motivations for presenting the current work. Unlike TM-AE, the methods outlined in Table 2 are capable of generating vectors for each word in the vocabulary, which can then be utilized to replace words and facilitate data augmentation. In contrast, TM-AE is restricted to generating embeddings for groups of words collectively, thereby limiting its applicability in this context.
> >
> > 5. **MLP Settings**
> > The following text has been added to Section 4.2:
> >
> > > Classification was performed using MLP classifier with default settings, including the random_state parameter set to 42, across all experiments and for all embedding sources.
> >
> > If necessary, a comprehensive list of default settings can be included in the appendix. These settings are consistent with those specified in the original code library, which is publicly accessible:
> >
> > - `hidden_layer_sizes=(100,)`
> > - `activation='relu'`
> > - `solver='adam'`
> > - `alpha=0.0001`
> > - `batch_size='auto'`
> > - `learning_rate='constant'`
> > - `learning_rate_init=0.001`
> > - `power_t=0.5`
> > - `max_iter=200`
> > - `shuffle=True`
> > - `random_state=None`
> > - `tol=0.0001`
> > - `warm_start=False`
> > - `momentum=0.9`
> > - `nesterovs_momentum=True`
> > - `early_stopping=False`
> > - `validation_fraction=0.1`
> > - `beta_1=0.9`
> > - `beta_2=0.999`
> > - `epsilon=1e-08`
> > - `n_iter_no_change=10`
> > - `max_fun=15000`

---

### Meta-Review · Area_Chair_as6S · 2024-12-21

**Metareview:**

The paper presents a two phase method for producing word embeddings using the Tsetlin Machine (TM) and the Coalesced Tsetlin Machine (CoTM) architectures.  The strengths of the paper lie in novel methodology and good writing that explains this carefully.  However, reviewers expressed several concerns including a lack of analysis of its computational efficiency (concerns), concerns about certain choices of experiments, including whether the two phase method is needed over a single phase one, and so forth.  This suggests that the paper can be improved in the future.

**Additional Comments On Reviewer Discussion:**

There was some discussion between reviewer aGqt and the authors that was productive.  Overall, the concern were not fully allayed.  There was a response from authors to the two other reviewers buy no further back and forth.

---

### Decision · Program_Chairs · 2025-01-22

Reject